# Statistical examination of shared loci in neuropsychiatric diseases using genome-wide association study summary statistics

Thomas P Spargo[1,2,3]*, Lachlan Gilchrist[1,4,5], Guy P Hunt[2,5,6], Richard JB Dobson[2,3,7,8], Petroula Proitsi[1], Ammar Al-Chalabi[1,9], Oliver Pain[1†], Alfredo Iacoangeli[1,2,3]*†

[1]Department of Basic and Clinical Neuroscience, Maurice Wohl Clinical Neuroscience Institute, King's College London, London, United Kingdom; [2]Department of Biostatistics and Health Informatics, King's College London, London, United Kingdom; [3]NIHR Maudsley Biomedical Research Centre (BRC) at South London and Maudsley NHS Foundation Trust and King's College London, London, United Kingdom; [4]Social, Genetic and Developmental Psychiatry Centre, Institute of Psychiatry, Psychology and Neuroscience, King's College London, London, United Kingdom; [5]Perron Institute for Neurological and Translational Science, Nedlands, Australia; [6]Centre for Molecular Medicine and Innovative Therapeutics, Murdoch University, Murdoch, Australia; [7]Institute of Health Informatics, University College London, London, United Kingdom; [8]NIHR Biomedical Research Centre at University College London Hospitals NHS21 Foundation Trust, London, United Kingdom; [9]King's College Hospital, London, United Kingdom

*For correspondence:
thomas.spargo@kcl.ac.uk (TPS);
alfredo.iacoangeli@kcl.ac.uk (AI)

†These authors contributed equally to this work

Competing interest: The authors declare that no competing interests exist.

## eLife assessment

This paper presents a **valuable** pipeline based on state-of-the-art analytical software that was used to study genetic pleiotropy between neuropsychiatric disorders. The presented evidence supporting the claims is **convincing** and now includes an appropriate comparison to previously published methods as well as a detailed exploration of the findings. The created pipeline can thus be used by researchers from diverse fields to study different combinations of diseases and traits.

**Abstract** Continued methodological advances have enabled numerous statistical approaches for the analysis of summary statistics from genome-wide association studies. Genetic correlation analysis within specific regions enables a new strategy for identifying pleiotropy. Genomic regions with significant 'local' genetic correlations can be investigated further using state-of-the-art methodologies for statistical fine-mapping and variant colocalisation. We explored the utility of a genome-wide local genetic correlation analysis approach for identifying genetic overlaps between the candidate neuropsychiatric disorders, Alzheimer's disease (AD), amyotrophic lateral sclerosis (ALS), frontotemporal dementia, Parkinson's disease, and schizophrenia. The correlation analysis identified several associations between traits, the majority of which were loci in the human leukocyte antigen region. Colocalisation analysis suggested that disease-implicated variants in these loci often differ between traits and, in one locus, indicated a shared causal variant between ALS and AD. Our study identified candidate loci that might play a role in multiple neuropsychiatric diseases and suggested the role of distinct mechanisms across diseases despite shared loci. The fine-mapping and colocalisation analysis protocol designed for this study has been implemented in a flexible analysis pipeline that produces HTML reports and is available at: https://github.com/ThomasPSpargo/COLOC-reporter.

## Introduction

The genetic spectrum of neuropsychiatric disease is diverse and various overlaps exist between traits. For instance, genetic pleiotropy between amyotrophic lateral sclerosis (ALS) and frontotemporal dementia (FTD) is increasingly recognised, and ALS is genetically correlated with Alzheimer's disease (AD), Parkinson's disease (PD), and schizophrenia (*Van Rheenen et al., 2021*; *Li et al., 2021*; *Ranganathan et al., 2020*). Improving understanding of the genetic architecture underlying these complex diseases could facilitate future treatment discovery.

Advances in genomic research techniques have accelerated discovery of genetic variation associated with complex traits. Genome-wide association studies (GWAS), in particular, have enabled population-scale investigations of the genetic basis of human diseases and anthropometric measures (*Abdellaoui et al., 2023*). Summary-level results from GWAS are being shared alongside publications with increasing frequency over time (*Reales and Wallace, 2023*), and a breadth of approaches now exist for downstream analysis based on summary statistics which can enable their interpretation and provide further biological insight.

Genetic correlation analysis allows estimation of genetic overlap between traits (*Bulik-Sullivan et al., 2015*; *Werme et al., 2022*; *Zhang et al., 2021*; *Shi et al., 2017*). A 'global' genetic correlation approach gives a genome-wide average estimate of this overlap. However, genetic relationships between traits can be obscured when correlations in opposing directions cancel out genome-wide (*Werme et al., 2022*). Recent methods allow for a more nuanced analysis, of 'local' genetic correlations partitioned across the genome (*Werme et al., 2022*; *Zhang et al., 2021*). This stratified approach to genome-wide analysis could prove effective for identifying pleiotropic regions and designing subsequent analyses aiming to identify genetic variation shared between traits.

A number of methods aim to disentangle causality within associated regions. This is important because the focus on single-nucleotide polymorphisms (SNPs), which are markers of genetic variation, in GWAS produces results that can be difficult to interpret, and causal variants are typically unclear. More so, because of linkage disequilibrium (LD), GWAS associations often comprise large sets of highly correlated SNPs spanning large genomic regions. Statistical fine-mapping is a common approach for dissecting complex LD structures and finding variants with implications for a given trait among the tens or hundreds that might be associated in the region (*Zou et al., 2022*).

Interpretation of regions associated with multiple traits can also be challenging, since it is often unclear whether these overlaps are driven by the same causal variant. Statistical colocalisation analysis can disentangle association signals across traits to suggest whether the overlaps result from shared or distinct causal genetic factors (*Wallace, 2021*; *Giambartolomei et al., 2018*; *Foley et al., 2021*). Traditionally, this analysis was restricted by the assumption of at most one causal variant for each trait in the region. However, recent extensions to the method now permit analysis based on univariate fine-mapping results for the traits compared and, therefore, analysis of regions with multiple causal variants.

Accordingly, we conducted genome-wide local genetic correlation analysis across five neuropsychiatric traits with recognised phenotypic and genetic overlap (*Li et al., 2021*; *Ranganathan et al., 2020*; *Ferrari et al., 2017*; *Weintraub and Mamikonyan, 2019 Beck et al., 2013*): AD, ALS, FTD, PD, and schizophrenia. Although several previous studies have performed global genetic correlation analyses between various combinations of these traits (*Van Rheenen et al., 2021*; *Li et al., 2021*; *Wainberg et al., 2023*; *McLaughlin et al., 2017*), we believe that this is the first to compare them at a genome-wide scale using a local genetic correlation approach. Loci highly correlated between trait pairs were further investigated with univariate fine-mapping and bivariate colocalisation techniques to examine variants driving these associations.

## Materials and methods
### Sampled GWAS summary statistics

We leveraged publicly accessible summary statistics from European ancestry GWAS meta-analyses of risk for AD (*Kunkle et al., 2019*), ALS (*Van Rheenen et al., 2021*), FTD (*Ferrari et al., 2014*), PD (*Nalls et al., 2019*), and schizophrenia (*Trubetskoy et al., 2022*). European ancestry data were selected to avoid LD mismatch between the GWAS sample and reference data from an external European population.

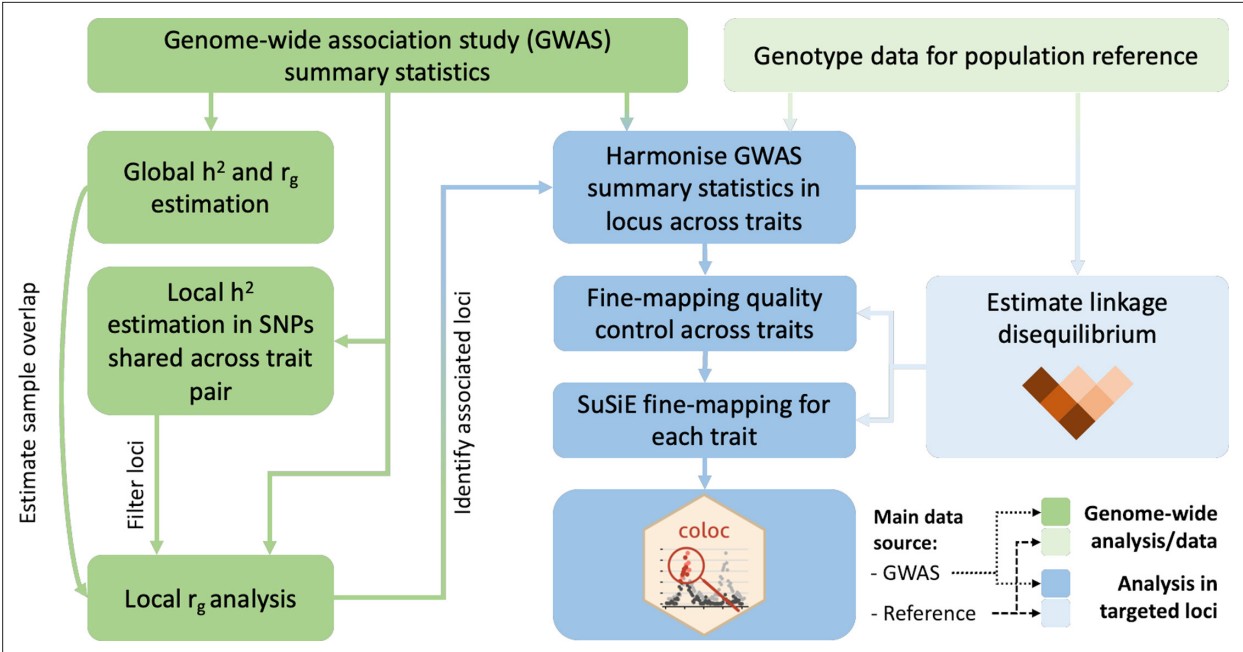

**Figure 1.** Overview of the analysis procedure for this study. SuSiE (sum of single effects) is a univariate fine-mapping approach implemented within the R package susieR. 'coloc' is an R package for bivariate colocalisation analysis between pairs of traits. $h^2$ = heritability, $r_g$ = bivariate genetic correlation. The analysis steps shaded in blue have been implemented within a readily applied analysis pipeline available on GitHub, copy archived at *Spargo, 2023*.

## Procedure

*Figure 1* summarises the analysis protocol for this study; further details are provided below.

## Processing of GWAS summary statistics

A standard data cleaning protocol was applied to each set of summary statistics (*Pain et al., 2021*). We retained only SNPs, excluding any non-SNP or strand-ambiguous variants. SNPs were filtered to those present within the 1000 Genomes phase 3 (1KG) European ancestry population reference dataset (*Auton et al., 2015*) ($N$ = 503). They were matched to the 1KG reference panel by GRCh37 chromosomal position using *bigsnpr* (version 1.11.6) (*Privé et al., 2018*), harmonising allele order with the reference and assigning SNP IDs.

If not reported, and where possible, effective sample size ($N_{eff}$) was calculated from per-SNP case and control sample sizes. When this could not be determined per-SNP, all variants were assigned a single $N_{eff}$, calculated as a sum of $N_{eff}$ values for each cohort within the GWAS meta-analysis (*Grotzinger et al., 2023*).

Further processing was performed where possible, excluding SNPs with p-values ≤0 or >1 and $N_{eff}$ >3 standard deviations from the median $N_{eff}$. Imputation INFO scores indicate the probability of each genotype given the available data and reference panel - we removed variants with INFO <0.9 if indicated. We filtered to include only variants with minor allele frequency (MAF) ≥0.005 in both the reference and GWAS samples and excluded SNPs with an absolute MAF difference of >0.2 between the two.

## Genome-wide analyses

### Global heritability and genetic correlations

LDSC (version 1.0.1) (*Bulik-Sullivan et al., 2015*; *Bulik-Sullivan et al., 2015*) was applied to estimate genome-wide univariate heritability ($h^2$) for each trait on the liability scale. The software was also applied to derive 'global' (i.e., genome-wide) genetic correlation estimates between trait pairs and estimate sample overlap from the bivariate intercept. The latter of these outputs was taken forward as an input for the local genetic correlation analysis using LAVA (see 2.2.2.2). Since global genetic

correlation analysis across the traits studied here is not novel and associations reported in past studies are congruent across different tools (*Wainberg et al., 2023*), the compatibility between LDSC and LAVA motivated our use of LDSC for this analysis.

These analyses were performed using the HapMap3 (*Altshuler et al., 2010*) SNPs and the LD score files provided with the software, calculated in the 1KG European population. No further MAF filter was applied (therefore variants with MAF ≥0.005 were included) and the other settings were left to their defaults.

## Local genetic correlation analysis

LAVA (version 0.1.0) (*Werme et al., 2022*) was applied to obtain local genetic correlation estimates across 2495 approximately independent blocks delineating the genome, based on patterns in LD. We used the blocks provided alongside the LAVA software which were derived from the 1KG European cohort. Bivariate intercepts from LDSC were provided to LAVA to estimate sample overlap between trait pairs.

LAVA was the most appropriate local genetic correlation approach for this study for several reasons (*Werme et al., 2022*). First, unlike SUPERGNOVA (*Zhang et al., 2021*) and rho-HESS (*Shi et al., 2017*), LAVA makes specific accommodations for analysis of binary traits. Second, other tools focus on bivariate correlation between traits while LAVA offers this alongside multivariate tests such as multiple regression and partial correlation, enabling rigorous testing of pleiotropic effects. Lastly, LAVA is shown to provide results which are less biased than those from other tools.

In accordance with prior studies, genetic correlation analysis was performed following an initial filtering step. Univariate heritability was estimated for each genomic block across SNPs in-common between a pair of traits, and only loci with local $h^2$ p-values below a threshold of $2.004 \times 10^{-5}$ (0.05/2495) in both traits continued to the bivariate analysis. This step ensures that univariate heritability is sufficient in both traits for a robust correlation estimate.

## Targeted genetic analyses

### Fine-mapping and colocalisation analysis

Statistical fine-mapping and colocalisation techniques were applied to further analyse associations between trait pairs in regions where the false discovery rate (FDR) adjusted p-value of local genetic correlation analysis was below 0.05 (after adjusting for all bivariate comparisons performed). Additional analysis was conducted at loci where significant correlations occurred between two trait pairs but not between the final pairwise comparison across the three implicated traits.

Fine-mapping was performed with *susieR* (v0.12.27) (*Zou et al., 2022*; *Wang et al., 2020a*), which implements the 'sum of single effects' (SuSiE) model to represent statistical evidence of causal genetic variation within 'credible sets' and per-SNP posterior inclusion probabilities (PIPs). A 95% credible set indicates 95% certainty that at least one SNP included within the set has a causal association with the phenotype and higher PIPs indicate a greater posterior probability of being a causal variant within a credible set. Multiple credible sets are identified when the data suggest more than one independent causal signal.

Colocalisation analysis was implemented with *coloc* (v5.1.0.1) (*Wallace, 2021*; *Giambartolomei et al., 2018*; *Giambartolomei et al., 2014*), which calculates posterior probabilities that a causal variant exists for neither, one, or both of two compared traits, testing also whether evidence for a causal variant in both traits suggests a shared variant (i.e., hypothesis 4 (H4); colocalisation) or independent signals (hypothesis 3 (H3)). Colocalisation analyses can be performed across all variants sampled in a region, under an assumption of at most one variant implicated per trait. It can also be performed using variants attributed to pairs of credible sets from SuSiE, relaxing the single variant assumption (*Wallace, 2021*). When evidence of a shared variant is found, the individual SNPs with the highest posterior probability of being that variant can be assessed. With a 95% confidence threshold, these are termed 95% credible SNPs.

## Analysis pipeline

We conducted colocalisation and fine-mapping analysis within an open-access pipeline developed for this study using R (v4.2.2) (*R Development Core Team, 2021*): https://github.com/ThomasPSpargo/COLOC-reporter, copy archived at *Spargo, 2023*.

Briefly, in this workflow (see *Figure 1*), GWAS summary statistics are harmonised across analysed traits for a specified genomic region, including only variants in common between them and available within a reference population. An LD correlation matrix across sampled variants is derived from a reference population using PLINK (v1.90) (*Purcell et al., 2007*; *Purcell, 2009*).

Quality control is performed per-dataset prior to univariate fine-mapping analysis. Diagnostic tools provided with *susieR* are applied to test for consistency between the LD matrix and *Z*-scores from the GWAS and identify variants with a potential 'allele flip' (reversed effect estimate encoding) that can impact fine-mapping.

Fine-mapping is performed for each dataset with the *coloc* package *runsusie* function, which wraps around *susie_rss* from *susieR* and is configured to facilitate subsequent colocalisation analysis. Sample size ($N_{eff}$ for binary traits) is specified as the median for SNPs analysed. Colocalisation analysis can be performed with the coloc functions *coloc.abf* and *coloc.susie* when fine-mapping yields at least one credible set for both traits and otherwise using *coloc.abf* only. Genes located near credible sets from fine-mapping and credible SNPs from colocalisation analyses are identified via Ensembl and *biomaRt (v2.54.0)* (*Durinck et al., 2005*; *Durinck et al., 2009*; *Cunningham et al., 2022*).

Analysis parameters can be adjusted by the user in accordance with their needs. Various utilities are included to help interpretation of fine-mapping and colocalisation results, including identification of genes nearby to putatively causal signals, HTML reports to summarise completed analyses, and figures to visualise the results and compare the examined traits.

## Current implementation

In this study, LD correlation matrices were derived from the 1KG European cohort. SNPs flagged for potential allele flip issues in either of the compared traits were removed from the analysis. Fine-mapping was performed with the *susie_rss* refine=TRUE option to avoid local maxima during convergence of the algorithm, leaving the other settings to the *runsusie* defaults. Colocalisation analysis was performed using the default priors for *coloc.susie* ($p_1 = 1 \times 10^{-4}$, $p_2 = 1 \times 10^{-4}$, $p_{12} = 5 \times 10^{-6}$).

Colocalisation and fine-mapping analyses were performed initially using the genomic blocks defined by LAVA, since these aim to define relatively independent LD partitions across the genome (*Werme et al., 2022*). If a 95% credible set could not be identified in one or both traits, we inspected local Manhattan plots for the region to determine whether potentially relevant signals occurred around the region boundaries. The analysis was repeated with a ±10 kb window around the LAVA-defined genomic region if p-values for SNPs at the edge of the block were $p < 1 \times 10^{-4}$ for both traits and the Manhattan plots were suggestive of a 'peak' not represented within the original boundaries.

**Table 1.** Genome-wide association studies (GWAS) sampled.
Each GWAS is a GWAS meta-analysis of disease risk across people of European ancestry.

| Trait | Estimated lifetime risk in population | GWAS Reference | N cases | N controls | Liability scale $h^2$ (standard error) |
|---|---|---|---|---|---|
| Alzheimer's disease | 1/10, *Chêne et al., 2015*[†] | *Kunkle et al., 2019* | 21,982 | 41,944 | 0.093 (0.0155) |
| Amyotrophic lateral sclerosis | 1/350, *Alonso et al., 2009*; *Johnston et al., 2006* | *Van Rheenen et al., 2021* | 27,205 | 110,881 | 0.0277 (0.003) |
| Frontotemporal dementia | 1/742, *Coyle-Gilchrist et al., 2016* | *Ferrari et al., 2014* | 2154 | 4308 | 0.0329 (0.0283) |
| Parkinson's disease | 1/37, *Parkinson's, 2017* | *Nalls et al., 2019* | 15,056 (+18,618 proxies*) | 449,056 | 0.0506 (0.0046) |
| Schizophrenia | 1/250, *Saha et al., 2005* | *Trubetskoy et al., 2022* | 53,386 | 77,258 | 0.1761 (0.0061) |

*Proxy cases from the UK Biobank Cohort.

[†]Estimated from cumulative risk after age 45 after correcting for competing risk of mortality and assuming a lifespan of ~85 years. $h^2$ = heritability.

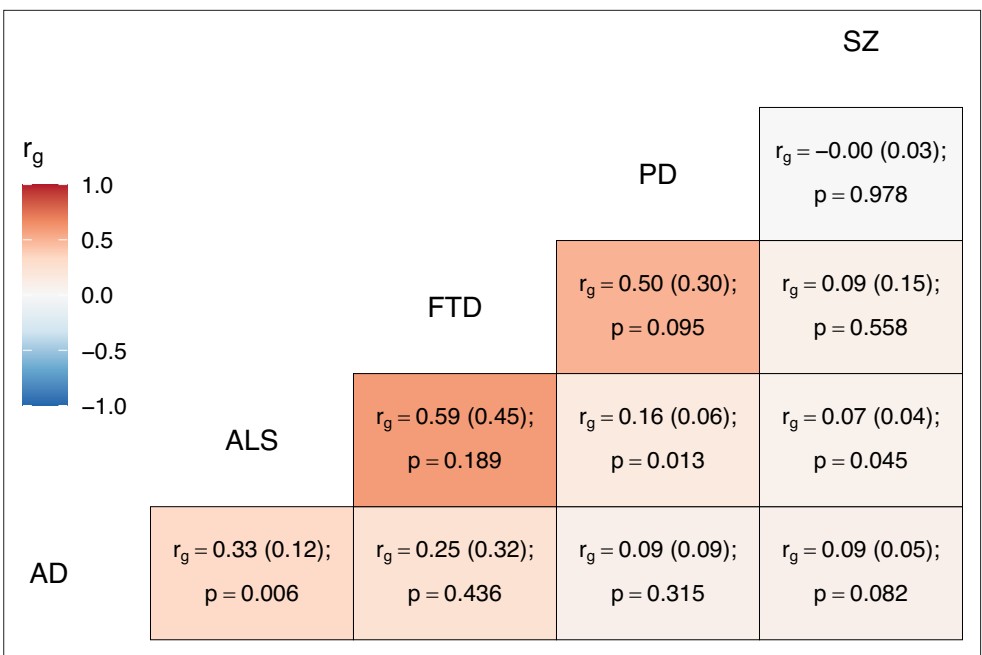

**Figure 2.** Genome-wide genetic correlation estimates between all trait pairs. The heatmap displays genetic correlations ($r_g$) each tile is labelled with the $r_g$ estimate and with its standard error in parentheses, alongside the p-value. AD = Alzheimer's disease, ALS = amyotrophic lateral sclerosis, FTD = frontotemporal dementia, PD = Parkinson's disease, SZ = schizophrenia.

## Results

### Genome-wide analyses

Descriptive information and heritability estimates for the sampled traits and GWAS are presented in *Table 1*. ALS had nominally significant global genetic correlations with schizophrenia (p = 0.045), PD (p = 0.013), and AD (p = 0.006); no other bivariate genome-wide correlations were statistically significant (see *Figure 2*).

A total of 605 local genetic correlation analyses were performed across all trait pairs in genomic regions where both traits passed the univariate heritability filtering step after restricting to SNPs sampled in both GWAS (see *Table 2*; *Figure 3*; *Supplementary file 1a*). The number of loci passing to bivariate analysis varied greatly across trait pairs and was congruent with the genome-wide heritability estimates (and their uncertainty) for each trait, reflecting differences in phenotypic variance explained by measured genetic variants and statistical power for each GWAS (see *Table 1*).

**Table 2.** Comparison of genome-wide SNP significance against local genetic correlation significance thresholds in all trait pairs and loci analysed.

All loci analysed showed sufficient local univariate heritability across compared traits to allow bivariate correlation analysis. Subsequent fine-mapping and colocalisation analyses were performed in this study for regions with at least a false discovery rate (FDR) adjusted significance for the local genetic correlation. SNP = single-nucleotide polymorphism.

| Number of traits in pair with genome-wide significant (p < 5 × 10⁻⁸) SNP in locus | Smallest significance threshold for local genetic correlation | | | |
|---|---|---|---|---|
| | Bonferroni (p < 8.26 × 10⁻⁵; 0.05/605) | FDR ($p_{fdr}$ < 0.05) | Nominal (p < 0.05) | Non-significant (p ≥ 0.05) |
| 0 | 1 | 17 | 77 | 394 |
| 1 | 1 | 4 | 18 | 80 |
| 2 | 0 | 3 | 2 | 8 |

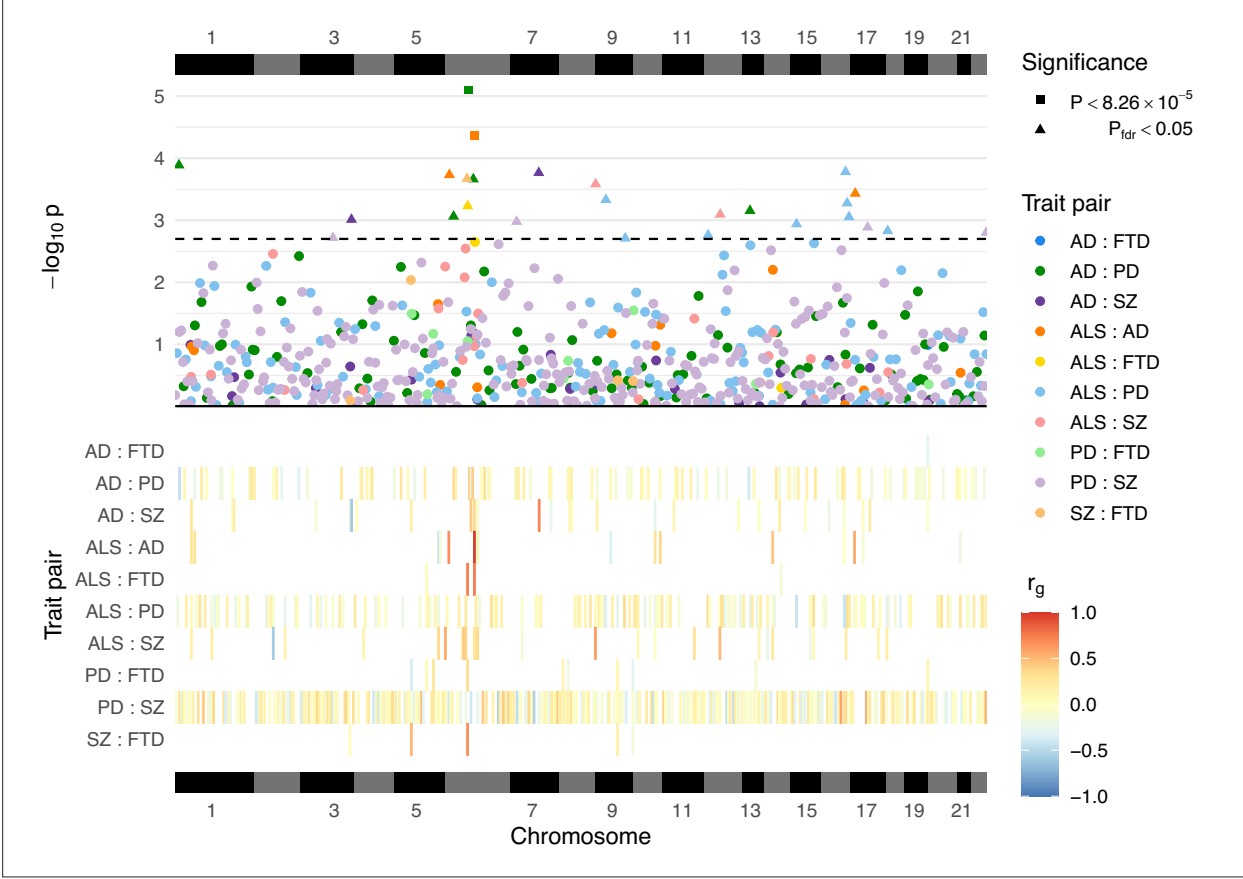

**Figure 3.** Local genetic correlation analyses between trait pairs. The lower panel displays a heatmap of genetic correlations ($r_g$) across genomic regions where any bivariate analyses were performed; white colouring indicates that the region was not analysed for a given trait pair owing to insufficient univariate heritability in one or both traits. The upper panel shows a Manhattan plot of p-values from each correlation analysis, denoting trait pairs by colour and comparisons passing defined significance thresholds by shape (square for a strict Bonferroni threshold and triangle for a false discovery rate [FDR] adjusted threshold); the hatched line indicates the threshold p-value above which $p_{fdr} < 0.05$. The panels are both ordered by relative genomic position, with bars above and below indicating each chromosome. AD = Alzheimer's disease, ALS = amyotrophic lateral sclerosis, FTD = frontotemporal dementia, PD = Parkinson's disease, SZ = schizophrenia. *Supplementary file 1a* provides a complete summary of local genetic correlation analyses performed. Twenty-six bivariate comparisons were significant following FDR adjustment ($p_{fdr} < 0.05$), two of which also passed the stringent Bonferroni threshold ($p < 8.26 \times 10^{-5}$; 0.05/605). While some regions included genome-wide significant single-nucleotide polymorphisms (SNPs) ($p < 5 \times 10^{-8}$) for one or both traits, others occurred in regions where genome-wide association studies (GWAS) associations were weaker (see *Table 2*). Five of these associations occurred at loci within the human leukocyte antigen (HLA) region (GRCh37: Chr6:28.48–33.45 Mb; 6p22.1–21.3, *Genome Reference Consortium, 2023*), and all five traits were implicated in at least one of these.

The online version of this article includes the following figure supplement(s) for figure 3:

**Figure supplement 1.** Comparison of positively and negatively correlated genetic loci.

## Targeted genetic analyses

Univariate fine-mapping and bivariate colocalisation analyses were subsequently performed to test for variants jointly implicated between trait pairs in regions with local genetic correlation $p_{fdr} < 0.05$. The ALS and schizophrenia trait pair was additionally examined at Chr6:32.22–32.45 Mb because significant genetic correlations were found between ALS and FTD and between schizophrenia and FTD at this locus. The correlation between ALS and schizophrenia at this locus had not been analysed owing to insufficient univariate heritability for ALS after restricting to SNPs in common with the schizophrenia GWAS.

Fine-mapping identified at least one 95% credible set for each of the compared traits for 7 of the 27 comparisons performed (see *Table 3*), and for one trait only in a further 5 (see *Supplementary file 1b*; *Supplementary file 1c*). This analysis suggested two credible sets for schizophrenia in the Chr12:56.99–58.75 Mb locus, for AD in Chr6:32.45–32.54 Mb, and (only when harmonised to SNPs in

**Table 3.** Colocalisation analysis conducted across 95% credible sets identified during univariate fine-mapping of trait pairs.

N SNPs refer to the number of SNPs present for both traits and the 1000 Genomes reference panel in the region within colocalisation and fine-mapping analysis.

| Trait 1 | Trait 2 | Genomic position (GRCh37) | Local genetic correlation estimate (95% confidence interval) | Fine-mapping credible set for trait 1 | 2 | N SNPs | SNP with highest PIP for fine-mapping credible set (nearest gene; sense-strand base pair distance) Trait 1 | Trait 2 | H0 | H1 | H2 | H3 | H4 |
|---|---|---|---|---|---|---|---|---|---|---|---|---|---|
| AD | PD | Chr6:32576785–32639239† | 0.406 (0.197, 0.648) | 1 | 1 | 958 | rs9271247 (HLA-DQA1; +15,844) | rs3129751 (HLA-DQA1; +13,767) | <0.01 | <0.01 | <0.01 | **0.95** | 0.05 |
| ALS | AD | Chr6:32629240–32682213* | 0.974 (0.717, 1.000) | 1 | 1 | 475 | rs9275477‡ (MTCO3P1; +1260) | rs9275207 (MTCO3P1; +16,191) | <0.01 | <0.01 | <0.01 | 0.10 | **0.90** |
|  |  |  |  |  | 1 |  |  | rs1980493 (BTNL2; 0) | <0.01 | <0.01 | 0.01 | **0.99** | <0.01 |
| ALS | FTD | Chr6:32208902–32454577§ | 0.723 (0.370, 1.000) | 1 | 2 | 1709 | rs9268833 (HLA-DRB9; 0) | rs9767620 (HLA-DRB9; +1498) | <0.01 | <0.01 | 0.01 | **0.99** | <0.01 |
|  |  | Chr6:32208902–32454577§ | - | 1 | 1 | 1711 | rs9268833 (HLA-DRB9; 0) | rs9268219 (C6orf10; 0) | <0.01 | <0.01 | <0.01 | **0.98** | <0.01 |
|  |  |  |  |  | 1 |  |  | rs12814239 (LRP1; 0) | <0.01 | <0.01 | <0.01 | **1.00** | <0.01 |
| ALS | SZ | Chr12:56987106–58748139 | 0.506 (0.218, 0.807) | 1 | 2 | 2260 | rs113247976 (KIF5A; 0) | rs324017 (NAB2; 0) | <0.01 | <0.01 | <0.01 | **1.00** | <0.01 |
| PD | SZ | Chr17:43460501–44865832 | 0.595 (0.266, 0.950) | 1 | 1 | 2453 | rs58879558 (MAPT; 0) | rs62062288 (MAPT; 0) | <0.01 | <0.01 | <0.01 | **0.81** | 0.19 |
| SZ | FTD | Chr6:32208902–32454577§ | 0.669 (0.379, 0.990) | 1 | 1 | 1657 | rs9268219 (C6orf10; 0) | rs9268877 (HLA-DRB9; 0) | <0.01 | <0.01 | <0.01 | **1.00** | <0.01 |

*Indicates comparisons with genetic correlation analysis $p < 8.26 \times 10^{-5}$ (0.05/605).

†Denotes locus extended by ±10 kb for fine-mapping and colocalisation analysis.

‡Variant identified in colocalisation as having the highest posterior probability of being shared variant assuming hypothesis 4 is true (see **Figure 4**).

§Differences in fine-mapping solutions across trait pairs in the Chr6:32.21–32.45 Mb locus reflect differences in the SNPs retained after restricting to those in common between the compared genome-wide association studies (GWAS).

¶H0 = no causal variant for either trait, H1 = variant causal for trait 1, H2 = variant causal for trait 2, H3 = distinct causal variants for each trait, H4 = a shared causal variant between traits. PIP = posterior inclusion probability, AD = Alzheimer's disease, ALS = amyotrophic lateral sclerosis, FTD = frontotemporal dementia, PD = Parkinson's disease, SZ = schizophrenia.

common with the ALS GWAS) for FTD in Chr6:32.22–32.45 Mb (see **Supplementary file 1c**). Although both positive and negative local genetic correlations passed the FDR-adjusted significance threshold, we observed only positive local genetic correlations in loci where fine-mapping credible sets were identified for both traits in the pair. This reflects that the absolute correlation coefficients and variant associations from the analysed GWAS studies were generally stronger in the positively correlated loci (see **Figure 3—figure supplement 1**).

Colocalisation analyses performed across fine-mapping credible sets and across all SNPs in a region generally gave support to the equivalent hypothesis (**Table 3**; **Supplementary file 1b**). Moreover, comparisons suggesting a signal was present in one trait only were largely concordant with the identification of fine-mapping credible sets in only that trait (**Supplementary file 1b**). **Figure 4—figure supplement 1** compares per-SNP p-values across trait pairs for comparisons with evidence of a relevant signal in both traits. **Figure 4—figure supplement 2** shows patterns of LD across SNPs assigned to credible sets for these analyses.

Strong evidence was found for a shared variant between ALS and AD within the Human leukocyte antigen (HLA) region (posterior probability of shared variant = 0.9; see **Figure 4**). The 95% credible SNPs for this association were distributed around the MTCO3P1 pseudogene and rs9275477, the lead genome-wide significant SNP from the ALS GWAS in this region, had the highest posterior probability of being implicated in both traits. **Figure 4—figure supplement 3** presents sensitivity analysis showing that the result is robust to a range of values for the shared-variant hypothesis prior probability.

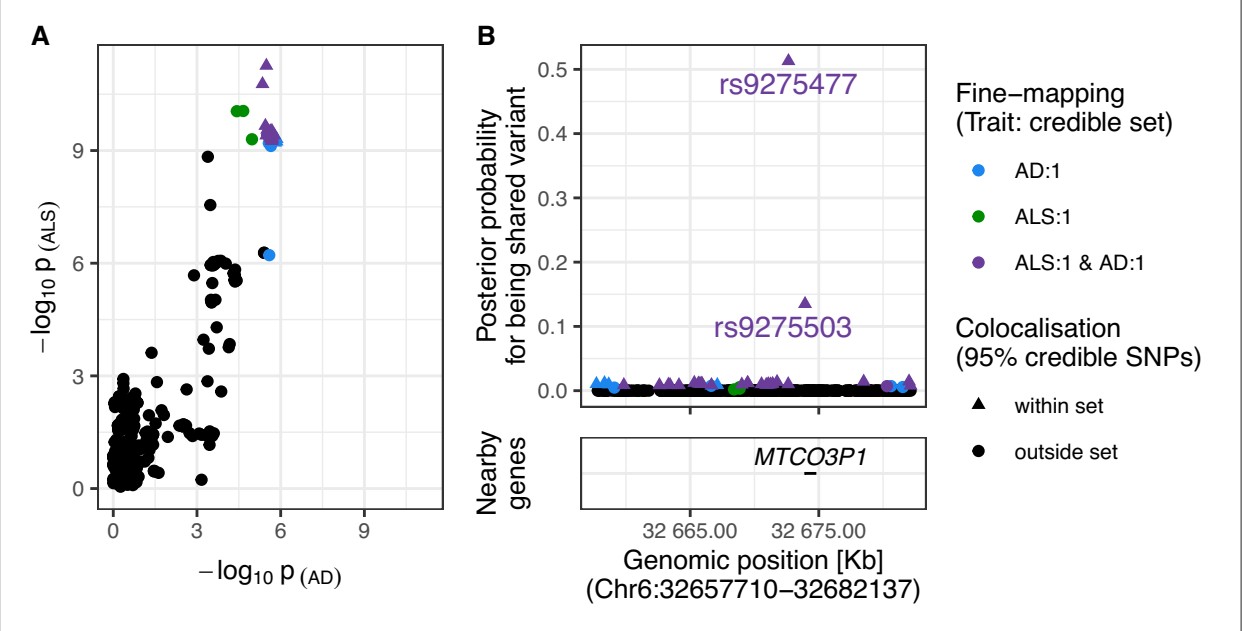

**Figure 4.** Evidence for colocalisation between amyotrophic lateral sclerosis (ALS) and Alzheimer's disease (AD) in the Chr6:32.63–32.68 Mb region. (**A**) Single-nucleotide polymorphism (SNP)-wise p-value distribution between ALS and AD across Chr6:32.63–32.68 Mb, in which colocalisation analysis found 0.90 posterior probability of the shared-variant hypothesis (see *Table 3*). (**B**) (upper) Per-SNP posterior probabilities for being a shared variant between ALS and AD, (lower) positions of HGNC gene symbols nearby to the 95% credible SNPs. Posterior probabilities for being a shared-variant sum to 1 across all SNPs analysed and are predicated on the assumption that a shared variant exists; 95% credible SNPs are those spanned by the top 0.95 of posterior probabilities. The *x*-axis for Panel B is truncated by the base pair range of the credible SNPs and genomic positions are based on GRCh37.

The online version of this article includes the following figure supplement(s) for figure 4:

**Figure supplement 1.** Single-nucleotide polymorphism (SNP)-wise p-value distribution between trait pairs in comparisons where colocalisation analysis suggested a causal variant in both traits.

**Figure supplement 2.** Heatmaps of linkage disequilibrium (LD) in the 1000 Genomes European reference population across variants assigned to any credible set during univariate fine-mapping of trait pairs (**A-G**).

**Figure supplement 3.** Sensitivity of colocalisation analysis to the prior probability of a shared variant between traits.

The other comparisons that found fine-mapping credible sets in both traits suggested that overlaps from the correlation analysis were driven by distinct causal variants (see *Table 3*; *Supplementary file 1b*).

Univariate fine-mapping of PD and schizophrenia at Chr17:43.46–44.87 Mb found large credible sets spanning many genes, including *MAPT* (*Allen et al., 2014*; *Snowden et al., 2015*; *Origone et al., 2018*; *Nakayama et al., 2019*; *Cheng et al., 2020*) and *CRHR1* (*Cheng et al., 2020*; *Bigdeli et al., 2021*) which have been previously implicated in the traits we have analysed. These expansive credible sets reflect the strong LD in the region and indicate a signal that is difficult to localise (see *Figure 4—figure supplement 2F*; *Supplementary file 1c*). The colocalisation analysis suggested independent variants for each trait despite many SNPs overlapping across their respective credible sets (see *Figure 4—figure supplement 2F*). Sensitivity analysis showed robust support for the two independent variants hypothesis across shared-variant hypothesis priors (*Figure 4—figure supplement 3*). However, the colocalisation analysis will increasingly favour the two independent variants hypothesis as the number of analysed variants increases (*Wallace, 2020*). Hence, the wide-spanning LD of this region may have obstructed identification of variants and mechanisms shared between the traits.

## Discussion

We examined genetic overlaps between the neuropsychiatric conditions AD, ALS, FTD, PD, and schizophrenia. Through genetic correlation analysis, we replicated genome-wide correlations previously

described between the studied traits (*Van Rheenen et al., 2021*; *Li et al., 2021*; *Wainberg et al., 2023*; *McLaughlin et al., 2017*). Leveraging a more recent local genetic correlation approach, we identified specific genomic loci jointly implicated between pairs of traits which were further investigated using statistical fine-mapping and colocalisation techniques.

Significant local genetic correlations were most frequent across genomic blocks within the *HLA* region, implicating each of the studied traits in at least one comparison. Several associated regions contained genes with known relevance for the traits studied, such as *KIF5A*, *MAPT*, and *CRHR1*. Colocalisation analysis found strong evidence for a shared genetic variant between ALS and AD in the Chr6:32.62–32.68 Mb locus within *HLA*, while the other colocalisation analyses suggested causal signals distinct across traits, for one trait only, or for neither trait.

The tendency for association between traits around the *HLA* region is reasonable, since this is a known hotspot for pleiotropy (*Werme et al., 2022*; *Watanabe et al., 2019*). The human leukocyte antigen (HLA) system is particularly known for its role in immune response and it is implicated in various types of disease (*Dendrou et al., 2018*; *Trowsdale and Knight, 2013*). Mounting evidence has linked *HLA* and associated genetic variation to the traits we have analysed, and mechanisms underlying these associations are beginning to be understood (*Dendrou et al., 2018*; *Trowsdale and Knight, 2013*; *Wang et al., 2020b*; *Song et al., 2016*; *Yu et al., 2021*; *Broce et al., 2018*; *Ferrari et al., 2016*; *Al-Diwani et al., 2017*; *Mokhtari and Lachman, 2016*; *Aliseychik et al., 2018*; *Zhang et al., 2022*). For instance, AD is associated with variants around the *HLA-DQA1* and *HLA-DRB1* genes and several SNPs in the non-coding region between them have been shown to modulate their expression (*Zhang et al., 2022*). Notably, one of the SNPs with a demonstrated regulatory role, rs9271247, had the highest probability of being causal for AD across the 95% credible set identified in the fine-mapping of the region.

Variants showing evidence for colocalisation between AD and ALS were distributed around the *MTCO3P1* pseudogene in the *HLA* class II non-coding region between *HLA-DQB1* and *HLA-DQB2*. *MTCO3P1* has been previously identified as one of the most pleiotropic genes in the GWAS catalog (*Chesmore et al., 2018*; *Sollis et al., 2023*). Previous studies have suggested the relevance of this region in both traits. *HLA-DQB1* and *HLA-DQB2* are both upregulated in the spinal cord of people with ALS, alongside other genes implicated in various immunological processes for antigen processing and inflammatory response (*Andrés-Benito et al., 2017*). *HLA* class II complexes, and their subcomponents, have been identified as upregulated in multiple brain regions of people with AD, using both gene and protein expression techniques (*Aliseychik et al., 2018*; *Hopperton et al., 2018*; *Pain et al., 2023*).

Our analysis of this region gave stronger support for colocalisation between the ALS and AD GWAS than a previous study (*Van Rheenen et al., 2021*). The previous study analysed a 200 kb window of over 2000 SNPs around the lead genome-wide significant SNP from the ALS GWAS, rs9275477, and found ~0.50 posterior probability for each of the shared and two independent variant(s) hypotheses. The current analysis used 475 SNPs occurring within a semi-independent LD block of ~50 kb in this locus. Since the posterior probability of the two independent variants hypothesis (H3) increases exponentially with the number of variants in the region while the shared-variant hypothesis (H4) scales linearly, it is expected that our analysis would give stronger support for the latter (*Wallace, 2020*). Given that the previous study defined regions for analysis based on an arbitrary window of ±100 kb around each lead genome-wide significant SNP from the ALS GWAS and we defined each analysis region based on patterns of LD in European ancestry populations, it is reasonable to favour the current finding.

More broadly, our analyses suggest that regions with a strong genetic correlation between the five traits studied often result from adjacent but trait-specific signals, likely reflecting overlaps between LD blocks (*Watanabe et al., 2019*). Correlations also occurred in regions with weaker overall GWAS associations (see *Table 2*), where fine-mapping and colocalisation analyses did not suggest causal associations in one or either trait. Such patterns likely reflect a shared polygenic trend across the region, rather than associations attributable to discrete variants. Accordingly, other approaches may be better suited for identifying regions containing genetic variation jointly causal across diseases, including the traditional approach of testing regions around overlapping genome-wide significant variants.

This study has used gold-standard statistical tools to examine genetic relationships between traits. The local genetic correlation analysis approach enabled targeted investigation of genomic regions

which appear to overlap between traits. The application of colocalisation analysis alongside a prior univariate fine-mapping step allowed for associations to be tested without conflating independent but nearby signals under the single-variant assumption of colocalisation analysis across all variants sampled in a region.

The study is not without limitation. We necessarily used the 1KG European reference population to estimate LD between SNPs. Fine-mapping is ideally performed with an LD matrix from the GWAS sample and is sensitive to misspecification when inconsistencies in LD occur between the reference and GWAS cohorts. Use of a reference population is not uncommon, and diagnostic tools available within the *susieR* package allow testing for inconsistencies between the reference and GWAS samples (*Zou et al., 2022*) . We accordingly implemented these tools centrally into our workflow and determined that the LD matrices from the 1KG reference were suitable for the data (estimates of *Z*-score and LD consistency are available in *Supplementary file 1c*). Nevertheless, repeating this study in under-represented populations would be an important future step to validate our findings.

We employed statistical methods to identify and analyse genomic regions containing variants which might be jointly implicated across traits. These approaches provide useful associations between traits identified from large-scale genomic datasets. However, they alone are not sufficient for translation into clinical practice. Future studies should aim to extend any associations found by integrating functional and multi-omics datasets to gain mechanistic insights into observed trends and facilitate treatment discovery (*Zhang et al., 2022*; *Pain et al., 2023*).

The fine-mapping and colocalisation analysis pipeline we have used is available as an open-access resource on GitHub to facilitate the application of these methods in future studies. Specified genomic regions can be readily analysed by providing GWAS summary statistics for binary or quantitative traits of interest and a population-appropriate reference dataset for estimation of LD. The pipeline returns resources including detailed reports that overview the analyses performed.

## Acknowledgements

The authors acknowledge the use of the CREATE research computing facility at *King's College London, 2022*. We also acknowledge Health Data Research UK, which is funded by the UK Medical Research Council, Engineering and Physical Sciences Research Council, Economic and Social Research Council, Department of Health and Social Care (United Kingdom), Chief Scientist Office of the Scottish Government Health and Social Care Directorates, Health and Social Care Research and Development Division (Welsh Government), Public Health Agency (Northern Ireland), British Heart Foundation and Wellcome Trust. This project was part funded by the MND Association and the Wellcome Trust. This is an EU Joint Programme-Neurodegenerative Disease Research (JPND) project. The project is supported through the following funding organisations under the aegis of JPND–http://www.neurodegenerationresearch.eu/ [United Kingdom, Medical Research Council (MR/L501529/1 and MR/R024804/1) and Economic and Social Research Council (ES/L008238/1)]. AAC is a NIHR Senior Investigator. AAC received salary support from the National Institute for Health Research (NIHR) Dementia Biomedical Research Unit at South London and Maudsley NHS Foundation Trust and King's College London. The work leading up to this publication was funded by the European Community's Health Seventh Framework Program (FP7/2007–2013; grant agreement number 259867) and Horizon 2020 Program (H2020-PHC-2014-two-stage; grant agreement number 633413). This project has received funding from the European Research Council (ERC) under the European Union's Horizon 2020 Research and Innovation Programme grant agreement no. 772376–EScORIAL. This study represents independent research part funded by the NIHR Maudsley Biomedical Research Centre at South London and Maudsley NHS Foundation Trust and King's College London. The views expressed are those of the author(s) and not necessarily those of the NHS, the NIHR, King's College London, or the Department of Health and Social Care. Funding was also provided by the King's College London DRIVE-Health Centre for Doctoral Training and the Perron Institute for Neurological and Translational Science. AI is funded by South London and Maudsley NHS Foundation Trust, MND Scotland, Motor Neurone Disease Association, National Institute for Health and Care Research, Spastic Paraplegia Foundation, Rosetrees Trust, Darby Rimmer MND Foundation, UK Research and Innovation (UKRI), Medical Research Council, LifeArc and Alzheimer's Research UK. OP is supported by a Sir Henry Wellcome Postdoctoral Fellowship [222811/Z/21/Z]. The funders had no role in study design, data collection and analysis, decision to publish, or preparation of the manuscript. This research was funded in

whole or in part by the Wellcome Trust [222811/Z/21/Z]. For the purpose of open access, the author has applied a CC-BY public copyright licence to any author accepted manuscript version arising from this submission.

## Additional information

### Funding

| Funder | Grant reference number | Author |
|---|---|---|
| Motor Neurone Disease Association | | Thomas P Spargo<br>Ammar Al-Chalabi<br>Alfredo Iacoangeli |
| Wellcome Trust | | Ammar Al-Chalabi<br>Oliver Pain |
| Medical Research Council | | Ammar Al-Chalabi<br>Alfredo Iacoangeli |
| Rosetrees Trust | | Alfredo Iacoangeli |
| MND Scotland | | Ammar Al-Chalabi<br>Alfredo Iacoangeli |
| Spastic Paraplegia Foundation | | Ammar Al-Chalabi<br>Alfredo Iacoangeli |
| Alzheimer's Research UK | | Ammar Al-Chalabi<br>Alfredo Iacoangeli |
| Darby Rimmer MND Foundation | | Alfredo Iacoangeli |
| NIHR Maudsley Biomedical Research Centre | | Thomas P Spargo<br>Richard JB Dobson<br>Ammar Al-Chalabi<br>Alfredo Iacoangeli |
| LifeArc | | Ammar Al-Chalabi<br>Alfredo Iacoangeli |
| Medical Research Council | MR/L501529/1 | Ammar Al-Chalabi |
| Medical Research Council | MR/R024804/1 | Ammar Al-Chalabi |
| Economic and Social Research Council | ES/L008238/1 | Ammar Al-Chalabi |
| European Community's Health Seventh Framework Programme | 259867 | Ammar Al-Chalabi |
| Horizon 2020 | 633413 | Ammar Al-Chalabi |
| Horizon 2020 | 772376–EScORIAL | Ammar Al-Chalabi |
| Wellcome Trust | 10.35802/222811 | Oliver Pain |
| King's College London | DRIVE-Health Centre for Doctoral Training | Lachlan Gilchrist |
| Perron Institute for Neurological and Translational Science | | Alfredo Iacoangeli |
| UK Research and Innovation | | Alfredo Iacoangeli |
| National Institute for Health and Care Research | | Alfredo Iacoangeli |

| Funder | Grant reference number | Author |
|---|---|---|
| South London and Maudsley NHS Foundation Trust | | Alfredo Iacoangeli |

The funders had no role in study design, data collection, and interpretation, or the decision to submit the work for publication. For the purpose of Open Access, the authors have applied a CC BY public copyright license to any Author Accepted Manuscript version arising from this submission.

### Author contributions

Thomas P Spargo, Conceptualization, Software, Formal analysis, Investigation, Methodology, Writing – original draft, Writing – review and editing; Lachlan Gilchrist, Investigation, Writing – review and editing; Guy P Hunt, Conceptualization, Software, Formal analysis, Writing – review and editing; Richard JB Dobson, Petroula Proitsi, Ammar Al-Chalabi, Conceptualization, Writing – review and editing; Oliver Pain, Conceptualization, Supervision, Writing – review and editing; Alfredo Iacoangeli, Conceptualization, Supervision, Funding acquisition, Methodology, Writing – original draft, Project administration, Writing – review and editing

### Author ORCIDs

Thomas P Spargo ⓘ https://orcid.org/0000-0003-4297-6418
Alfredo Iacoangeli ⓘ https://orcid.org/0000-0002-5280-5017

Reviewer #1 (Public Review): https://doi.org/10.7554/eLife.88768.3.sa1
Reviewer #2 (Public Review): https://doi.org/10.7554/eLife.88768.3.sa2
Author response https://doi.org/10.7554/eLife.88768.3.sa3

## Additional files

### Supplementary files

• Supplementary file 1. Extended results from local genetic correlation analysis and subsequent fine-mapping and colocalisation analyses across loci correlated between trait pairs. (**a**) Results of all bivariate local genetic correlation analyses. (**b**) Results of colocalisation analyses performed across all single-nucleotide polymorphisms (SNPs) sampled in the region. Number of SNPs refers to the number of SNPs in common between the two traits analysed and present within the 1000 genomes reference panel. Comparisons where univariate fine-mapping identified at least one credible set in each trait were also performed on the basis of these credible sets (see *Table 3*). Annotations column: * denotes comparisons with genetic correlation analysis p-values below the strict Bonferroni correction threshold; all others passed false discovery rate (FDR) correction. Δ denotes locus with boundaries extended by ±10 kb compared to the region partition defined in genetic correlation analysis. (**c**) Overview of credible sets identified across fine-mapping analyses in summary statistics harmonised across trait pairs.

• MDAR checklist

### Data availability

All data used in this study are publicly available, including summary statistics from genome-wide association studies (GWAS) across each of the five analysed traits (*Van Rheenen et al., 2021*; *Kunkle et al., 2019*; *Ferrari et al., 2014*; *Nalls et al., 2019*; *Trubetskoy et al., 2022*; *Pain et al., 2021*) and the 1000 Genomes Phase 3 reference dataset (*Auton et al., 2015*). Code used to conduct the analyses described in this manuscript are available on GitHub, copy archived at *Spargo, 2023*.

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
