## [Editor Report · eLife assessment]

This paper presents a **valuable** pipeline based on state-of-the-art analytical software that was used to study genetic pleiotropy between neuropsychiatric disorders. The presented evidence supporting the claims is **convincing** and now includes an appropriate comparison to previously published methods as well as a detailed exploration of the findings. The created pipeline can thus be used by researchers from diverse fields to study different combinations of diseases and traits.

---

## [Referee Report · Reviewer #1 (Public Review)]

The authors investigate pleiotropy in the genetic loci previously associated to a range of neuropsychiatric disorders: Alzheimer's disease, amyotrophic lateral sclerosis (ALS), frontotemporal dementia, Parkinson's disease, and schizophrenia. The local statistical fine-mapping and variant colocalisation approaches they use have the potential to uncover not only shared loci but also shared causal variants between these disorders. There is existing literature describing the pleiotropy between ALS and these other disorders but here the authors apply state-of-the-art, local genetic correlation approaches to further refine any relationships.

Complex disease and GWAS is not my area of expertise but the authors managed to present their methods and results in a clear, easy-to-follow manner. Their results statistically support several correlations between the disorders and, for ALS and AD, a shared variant in the vicinity of the lead SNP from the original ALS GWAS. Such findings could have important implications for our understanding of the mechanisms of such disorders and eventually the possibility of managing and treating them.

The authors have built a useful pipeline that plugs together all the gold-standard, existing software to perform this analysis and made it openly available which is commendable. However, there is little discussion of what software is available to perform global and local correlation analysis and, if there are multiple tools available, why they consider the ones they selected to be the gold-standard.

There is some mention of previous findings of genetic pleiotropy between ALS and these other disorders in the introduction, and discussion of their improved ALS-AD evidence relative to previous work. However, detailed comparisons of their other correlations to what was described before for the same pairs of disorders (if any) is missing. Adding this would strengthen the impact of this paper.

Finally, being new to this approach I found the abstract a little confusing. Initially, the shared causal variant between ALS and AD is mentioned but immediately in the following sentence they describe how their study "suggested that disease- implicated variants in these loci often differ between traits". After reading the whole paper I understood that the ALS-AD shared variant was the exception but it may be best to restructure this part of the abstract. Additionally, in the abstract the authors state that different variants "suggests the role of distinct mechanisms across diseases despite shared loci". Is it not possible that different variants in the same regulatory region or protein-coding parts of a gene could be having the same effect and mechanism? Or does the methodology to establish that different variants are involved automatically mean that the variants are too distant for this to be possible?

These concerns were addressed in the revised version of this manuscript.

---

## [Referee Report · Reviewer #2 (Public Review)]

Summary:

Spargo and colleagues present an analysis of the shared genetic architectures of Schizoprehnia and several late-onset neurological disorders. In contrast to many polygenic traits for which global genetic correlation estimates are substantial, global genetic correlation estimates for neurological conditions are relatively small, likely for several reasons. One is that assortative mating, which will spuriously inflate genetic correlation estimates, is likely to be less salient for late-onset conditions. Another, which the authors explore in the current manuscript, is that some loci affecting two or more conditions (i.e., pleiotropic loci) may have effects in opposite directions, or shared loci are sparse, such that the global genetic correlation signal washes out.

The authors apply a local genetic correlation approach that assesses the presence and direction of pleiotropy in much smaller spatial windows across the genome. Then, within regions evidencing local genetic correlations for a given trait pair, they apply fine-mapping and colocalization methods to attempt to differentiate between two scenarios: that the two traits share the same causal variant in the region or that distinct loci within the region influence the traits. Interestingly, the authors only discover one instance of the former: an SNP in the HLA region appearing to confer risk for both AD and ALS. This is in contrast to six regions with distinct causal loci, and twenty regions with no clear shared loci.

Finally, the authors have published their analysis pipeline such that other researchers might easily apply the same techniques to other collections of traits.

Strengths:

- All such analysis pipelines involve many decision points where there is often no clear correct option. Nonetheless, the authors clearly present their reasoning behind each such decision.

- The authors have published their analytic pipeline such that future researchers might easily replicate and extend their findings.

Weaknesses:

- The majority of regions display no clear candidate causal variants for the traits, whether shared or distinct. Further, despite the potential of local genetic correlation analysis to identify regions with effects in opposing directions, all of the regions for causal variants were identified for both traits evidenced positive correlations. The reasons for this aren't clear and the authors would do well to explore this in greater detail.

- The authors very briefly discuss how their findings differ from previous analyses because of their strict inclusion for "high-quality" variants. This might be the case, but the authors do not attempt to demonstrate this via simulation or otherwise, making it difficult to evaluate their explanation.

These concerns were addressed in the revised version of this manuscript.

---

## [Author Response]

The following is the authors’ response to the original reviews.

**Public Reviews:**

**Reviewer #1 (Public Review):**
The authors investigate pleiotropy in the genetic loci previously associated to a range of neuropsychiatric disorders: Alzheimer's disease, amyotrophic lateral sclerosis (ALS), frontotemporal dementia, Parkinson's disease, and schizophrenia. The local statistical fine-mapping and variant colocalisation approaches they use have the potential to uncover not only shared loci but also shared causal variants between these disorders. There is existing literature describing the pleiotropy between ALS and these other disorders but here the authors apply state of the art, local genetic correlation approaches to further refine any relationships.Complex disease and GWAS is not my area of expertise but the authors managed to present their methods and results in a clear, easy to follow manner. Their results statistically support several correlations between the disorders and, for ALS and AD, a shared variant in the vicinity of the lead SNP from the original ALS GWAS. Such findings could have important implications for our understanding of the mechanisms of such disorders and eventually the possibility of managing and treating them.The authors have built a useful pipeline that plugs together all the gold-standard, existing software to perform this analysis and made it openly available which is commendable. However, there is little discussion of what software is available to perform global and local correlation analysis and, if there are multiple tools available, why they consider the ones they selected to be the gold-standard.There is some mention of previous findings of genetic pleiotropy between ALS and these other disorders in the introduction, and discussion of their improved ALS-AD evidence relative to previous work. However, detailed comparisons of their other correlations to what was described before for the same pairs of disorders (if any) is missing. Adding this would strengthen the impact of this paper.Finally, being new to this approach I found the abstract a little confusing. Initially, the shared causal variant between ALS and AD is mentioned but immediately in the following sentence they describe how their study "suggested that disease- implicated variants in these loci often differ between traits". After reading the whole paper I understood that the ALS-AD shared variant was the exception but it may be best to restructure this part of the abstract. Additionally, in the abstract the authors state that different variants "suggests the role of distinct mechanisms across diseases despite shared loci". Is it not possible that different variants in the same regulatory region or protein-coding parts of a gene could be having the same effect and mechanism? Or does the methodology to establish that different variants are involved automatically mean that the variants are too distant for this to be possible?

We thank reviewer one for their considered review of this manuscript and for highlighting points that would benefit from further exploration. Itemised responses are provided below.

(1) The reviewer noted that we did not adequately explain our choice of software for global and local genetic correlation analysis, and why we consider the techniques chosen as gold standard. We agree that the paper would benefit from clarification around this aspect of the study.

Briefly, we firstly selected LAVA for the local genetic correlation analysis because it offers several advantages above competing software and was developed by a reputable team previously known for developing MAGMA, which is well-established in the statistical genetics field. In the manuscript (page 8), we added the following clarification: “LAVA was the most appropriate local genetic correlation approach for this study for several reasons. First, unlike SUPERGNOVA and rho-HESS, LAVA makes specific accommodations for analysis of binary traits. Second, other tools focus on bivariate correlation between traits whilst LAVA offers this alongside multivariate tests such as multiple regression and partial correlation, enabling rigorous testing of pleiotropic effects. Lastly, LAVA is shown to provide results which are less biased than those from other tools.”

LDSC was selected for the global genetic correlation analysis because the software is well-established and likely the most widely adopted global genetic correlation tool. Reflecting its prevalence, the software is also compatible with LAVA, which adjusts for sample overlap based on the bivariate intercept estimate returned by LDSC. Since global genetic correlations were not the primary focus of this study, having been tested across several previous investigations (see response 2), we did not prioritise comparison of correlation estimates from LDSC against other available software. In the manuscript (pages 7-8) we now include the following statement: “[LDSC] was also applied to derive ‘global’ (i.e., genome-wide) genetic correlation estimates between trait pairs and estimate sample overlap from the bivariate intercept. The latter of these outputs was taken forward as an input for the local genetic correlation analysis using LAVA (see 2.2.2.2). Since global genetic correlation analysis across the traits studied here is not novel and associations reported in past studies are congruent across different tools, the compatibility between LDSC and LAVA motivated our use of LDSC for this analysis”.

(2) The second comment was that the paper would be strengthened by contextualising our study with detail around what is previously known about associations between the studied traits. Accordingly, we have added clarifying text at the end of the introduction, stating: “although previous studies have performed global genetic correlation analyses between various combinations of these traits {references}, this is the first to compare them at a genome-wide scale using a local genetic correlation approach“. In the discussion, we link back to these studies, stating that “Through genetic correlation analysis, we replicated genome-wide correlations previously described between the studied traits {references}”.

(3) The reviewer highlighted that the abstract as originally written may mislead or confuse the reader and we agree that clarity could be improved with some restructuring. This has now been revised and should read more logically.

(4) They also enquired about our reasons for suggesting that the implication of distinct variants for each trait from a colocalisation analysis suggests a distinct causal mechanism. We thank them for this question as it encouraged us to reconsider how best to present the results of this analysis. To answer their question:

It is certainly true that nearby but distinct variants can confer the same effect. In a scenario where multiple distinct variants result in the same effect and thus increase susceptibility towards two or more related phenotypes, you would expect to find evidence of association to each relevant variant in GWAS across these related traits (even if the magnitude of the associations differ). Where biological mechanisms are shared, post-GWAS finemapping analysis would be expected to yield credible sets overlapping across the traits, and likewise, colocalisation analysis should converge on a set of credible SNPs that are candidates for the shared effect. Where multiple distinct variants confer the same effect, you would expect to see separate fine-mapping credible sets for these distinct variants that colocalise pairwise between the jointly-affected traits. Generally, therefore, evidence supporting the two distinct variants hypothesis would suggest the role of two distinct mechanisms except when certain credible sets identified through fine-mapping converge on a colocalised effect.

There is a further caveat which we also explored in response to Reviewer two: if a region includes long-spanning LD (and hence a larger number of variants are considered in the analysis), then the colocalisation analysis is more likely to favour the two distinct variants hypothesis since the probability of the variants implicated in both traits being shared decreases. It is likely that support for the two independent variants hypothesis is correct in most of the comparisons from this study that favour this conclusion. This is because, generally, the fine-mapping credible sets do not overlap across trait pairs (Figure S4) and consequently the colocalisation analysis does not find any support for the shared variant hypothesis. An exception is the analysis of PD and schizophrenia at the *MAPT* locus on chromosome 17. We have accordingly added the following clarification to the (page 18): “However, the colocalisation analysis will increasingly favour the two independent variants hypothesis as the number of analysed variants increases. Hence, the wide-spanning LD of this region may have obstructed identification of variants and mechanisms shared between the traits.”

**Reviewer #2 (Public Review):**
Summary:Spargo and colleagues present an analysis of the shared genetic architectures of Schizoprehnia and several late-onset neurological disorders. In contrast to many polygenic traits for which global genetic correlation estimates are substantial, global genetic correlation estimates for neurological conditions are relatively small, likely for several reasons. One is that assortative mating, which will spuriously inflate genetic correlation estimates, is likely to be less salient for late-onset conditions. Another, which the authors explore in the current manuscript, is that some loci affecting two or more conditions (i.e., pleiotropic loci) may have effects in opposite directions, or shared loci are sparse, such that the global genetic correlation signal washes out.The authors apply a local genetic correlation approach that assesses the presence and direction of pleiotropy in much smaller spatial windows across the genome. Then, within regions evidencing local genetic correlations for a given trait pair, they apply fine-mapping and colocalization methods to attempt to differentiate between two scenarios: that the two traits share the same causal variant in the region or that distinct loci within the region influence the traits. Interestingly, the authors only discover one instance of the former: an SNP in the HLA region appearing to confer risk for both AD and ALS. This is in contrast to six regions with distinct causal loci, and twenty regions with no clear shared loci.Finally, the authors have published their analysis pipeline such that other researchers might easily apply the same techniques to other collections of traits.Strengths:- All such analysis pipelines involve many decision points where there is often no clear correct option. Nonetheless, the authors clearly present their reasoning behind each such decision.- The authors have published their analytic pipeline such that future researchers might easily replicate and extend their findings.Weaknesses:- The majority of regions display no clear candidate causal variants for the traits, whether shared or distinct. Further, despite the potential of local genetic correlation analysis to identify regions with effects in opposing directions, all of the regions for causal variants were identified for both traits evidenced positive correlations. The reasons for this aren't clear and the authors would do well to explore this in greater detail.- The authors very briefly discuss how their findings differ from previous analyses because of their strict inclusion for "high-quality" variants. This might be the case, but the authors do not attempt to demonstrate this via simulation or otherwise, making it difficult to evaluate their explanation.

We thank Reviewer two for their appraisal of this manuscript and kind comments regarding its strengths. We will now aim to address the identified weaknesses.

(1) The reviewer comments that we did not adequately investigate why loci with causal variants identified in both traits all had positive local genetic correlations. We agree that it would be helpful to better understand the underlying reasons. To address this issue, we have added a new supplementary figure to compare the positive and negative local genetic correlation results (see Figure S2). In the main-text we add the following clarification. ”Although both positive and negative local genetic correlations passed the FDR-adjusted significance threshold, we observed only positive local genetic correlations in loci where fine-mapping credible sets were identified for both traits in the pair. This reflects that the correlation coefficients and variant associations from the analysed GWAS studies were generally stronger in the positively correlated loci (see Figure S2).”

(2) The reviewer rightly suggests that the manuscript would benefit from an improved explanation of the somewhat inconsistent results for the colocalisation analysis of ALS and AD at the locus around the rs9275477 SNP from this work and a previous study. We have now further investigated this and believe that the discrepancy results partly from an inherent empirical characteristic of the colocalisation analysis. We have explained this in the manuscript (page 22) as follows: “The previous study analysed a 200Kb window of over 2,000 SNPs around the lead genome-wide significant SNP from the ALS GWAS, rs9275477, and found ~0.50 posterior probability for each of the shared and two independent variant(s) hypotheses. The current analysis used 475 SNPs occurring within a semi-independent LD block of ~50kb in this locus. Since the posterior probability of the two independent variants hypothesis (H3) increases exponentially with the number of variants in the region whilst the shared variant hypothesis (H4) scales linearly, it is expected that our analysis would give stronger support for the latter. Given that the previous study defined regions for analysis based on an arbitrary window of ±100kb around each lead genome-wide significant SNP from the ALS GWAS and we defined each analysis region based on patterns of LD in European ancestry populations, it is reasonable to favour the current finding.”